# The Long Road Out of Eden: Early Dynastic Temples, a Quantitative Approach to the Bent-Axis Shrines

**Pascal Butterlin**

VEPMO, UMR 7041 ARSCAN Laboratory, Art Institute, Paris 1 Panthéon University, 75006 Paris, France;
pascal.butterlin@univ-paris1.fr

**Abstract:** This study was conducted to quantitatively assess the architectural data stemming from 70 buildings usually considered as bent-axis temples, a type of Mesopotamian temple mainly constructed from 2900 to 2300–2200 BC. The study reviews, region-by-region and site-by-site, the dimensions of the rooms considered the "holy of holies", registering width, length, and surface area. The results are discussed in comparison to the previous reception rooms of the tripartite buildings, considered the original matrix from which these shrines developed. The chronological and regional differences that are outlined provide some insights about the kind of social units that were involved in the use of those buildings, which were key structures in the urban fabric of Early City states.

**Keywords:** sacred architecture; Early City States; bent-axis temples

## 1. Introduction

Defining the relationships between the sacred and architecture in the ancient Near East has been one of the cornerstones of the studies of the development of ancient City-States. This architecture is directly linked to the idea that the roots of power lay in the sacred; that kings were, first of all, "king priests"; and that the birth of the state was actually linked to the institutionalization of religion. Architecture has been considered as one of the obvious material signatures of that process through the identification of non-domestic buildings as temples, usually characterized by their monumentality and the use of specific plans, decoration, or installations. For ancient Mesopotamia, the discussion has largely been centered on identifying the steps in the differentiation between religious and secular architecture, namely between temple and palace.

Even if debate is ongoing about the role of the sacred in palatial architecture, this difference was well-established at the end of the third millennium, at least in Mesopotamia. This is less the case for earlier periods, especially during the formative phase of the states, known as the proto-urban period, and the first half of the third millennium. Since the pioneering work of Andrae, the idea has prevailed of a continuous development of religious architecture from the original tripartite buildings toward a complex set of buildings arranged along a typology of sacred spaces and their immediate environment, dependencies, or courts. From this original matrix of tripartite or bipartite buildings, the temple, conceived either as a passage or a house, was organized along elementary forms (Andrae 1930), either as langraum or breitraum (Heinrich 1982). It became the center of a phenomenological approach of an almost atemporal model of the "Near Eastern temple". This is largely the case with the idea of the tripartition of the progression toward the sacred: enclosure of the temple, place of worship/sacrifice, and place of the epiphany (Parrot 1956; Margueron 1995). This type of approach has often been criticized for being ahistorical, so a new method of approaching data, namely architecture, ritual installations, and objects, has progressively developed.

In order to overcome these objections, one approach is the reassessment of the architectural data, especially for the formative phases: the Late Chalcolithic (Butterlin 2015) and

Early Dynastic periods in Greater Mesopotamia. One of the major challenges is to understand why tripartite architecture that was the matrix of the development of monumental architecture in Mesopotamia ceased almost completely to be the major pattern of organization of prestige architecture. It was replaced by different formulas, such as "temple-houses" or "bent-axis shrines", in antis temples or different types of monocellular/multicellular arrangements, from long to broad room buildings. Those elementary forms, with their different layouts, were assessed first and their architectural environment, usually courtyards, adjoining rooms, and various access systems, were then studied. Progressive or restricted access to the sacred has been investigated through the archaeology of movement, a dynamic approach focused less on static positions than on how these buildings and spaces were used and arranged with specific installations or ritual stages from the outside to the inside of the building, but also the streets.

Evaluating the development of monumental architecture, without quickly assigning labels such as temple, palace, or temple-palace, I propose to look at the evolution of what is usually considered as a "prestige room", either reception rooms, ceremonial rooms, throne rooms, or cult rooms. Usually considered as the holy of holies, they are places of specific staging; majesty staging, usually marked by a podium or altar; and different settings. These places were treated typologically; I propose a quantitative approach based on assessing the capacity of those rooms, appreciated through dimension, surface area, patterns of circulation, doors, and focal points. It will therefore be possible to discuss their accommodation capacity, as one criterion to discuss how they were used or at least conceived.

## 2. Results

In order to assess the documentation, I concentrated on these reception rooms, usually considered the most important room, that is, the reception room or *cella*. Usually distinguished through its size, its position in the whole building, and its specific arrangements, this kind of room is usually defined as the place of epiphany or the manifestation of power, religious or not.

To assess the data, I first looked at the dimensions of those rooms (length, width, and surface area) and then at the circulation patterns, which might describe how these rooms were used. This provides some insights into the kind of staging that was at work, a grammar of prestige and display. However, it also provides a glimpse into factual hierarchy that in no way prejudges the role and the function of the building. Second, to understand the whole dynamic at work, I introduce the idea of religiosity of movement, at different levels.

In a previous study, I presented 113 tripartite buildings and identified clear patterns in the evolution of those buildings in terms of scale and organization of domestic and monumental space (Butterlin 2018). This allowed the definition of some architectural modules, characteristic of certain tripartite buildings, notably the temples at Tepe Gawra, or what I propose to name the monumental standard of the LC 4-5 periods, at Uruk, or in the Urukean colonies. These formulas are responses to real needs in terms of scale, technical skills in coverage capacity, scenography, and room capacity. I add to those data, which is discussed below with some additions, a study of the temples of Southern, Central Mesopotamia, and Upper Mesopotamia, as they were defined in the Arcane Project, including the Diyala and Mari temples, in the discussion (Figure S1, inserted in supplementary material).

### 2.1. The Early Dynastic Temples of Mari

The Mari data was re-evaluated recently (Butterlin 2021a). To the famous temples excavated by Parrot, and later Margueron, we added new information from the archives of the Mari archaeological expedition and the results of the last excavations I directed upon the site from 2005 to 2010. So far, seven temples have been identified (Figure S2). Mari has the largest number of sanctuaries dating the middle of the third millennium BC. Thirty deities are mentioned in the cuneiform records of the second city of Mari (Lecompte 2021),

and even if some divinities are duplicates, this number indicates the number of sanctuaries expected to be recovered at the site.

Some of those temples, notably the Ishtar Ush Temple, the Enceinte Sacrée in the palace, and the recently discovered Temple of the Lord of the Land (Butterlin 2014), present at least three levels, with some important redevelopments. In two cases, apart from the holy of holies usually well-identified, two buildings present chapels or an additional cella. These are specific rooms with enough settings such as banquettes, which may indicate a secondary place of worship, or a second one. This is the case for the Ishtar Ush Temple (cella 18 from level a onward) and at the Ninhursag Temple (Figure S3), with the "chambre aux banquettes" (Beyer 2021a). All these buildings are generally considered bent-axis temples, although it is not absolutely certain that the podium was situated on the short side of the building. The inner organization of the cella 17 of the Ishtar Temple and of the cella of the Ninhursag Temple are both characterized by a low podium facing the entrance with installations on both sides. The place of the divine statue, if it existed, remains to be defined.

In total, we considered 10 rooms (Table 1 and Figure S4). They are all dated to ville II (between 2500 and 2300 BC), which corresponds to the Early Dynastic III a and b of the Diyala sequence and ECM 4:5 periods of the Arcane project (Reichel 2009). In the case of multilevel temples, we only considered it as a new item if the dimensions varied: cella 17 and cella 18 at Ishtar Ush (Parrot 1956; Margueron 2004, pp. 246–48; Margueron 2017; Beyer 2021a); the Ninhursag cella and chambre aux banquettes (Beyer 2014, Beyer 2021a), Ishtarat, room 5, Inanna Zaza, cella 13 (Parrot 1967; Margueron 2004, pp. 241–43), Shamash, cella (Margueron 2004, pp. 240–41); Temple of the Lord of the Land, cella of level II, cella of level III (Butterlin 2014; Beyer 2014, 2021a); and the sacred space XLVI of the Enceinte Sacrée, which does not vary in dimension from Palace P 3 to P 0 (Margueron 2004, pp. 210–14). Among these temples, the Temple of the Lord of the Land presents two altars on level II, facing the entrance of the cella (Figure S5), after a bent approach from the monumental entrance.

The length varies from 8 to 29 m (room XLVI of the Enceinte Sacrée). Most of the spaces range between 9 and 11 m in length. Three buildings distinguish themselves: the Inanna Zaza Temple, the Shamash Temple, and the Enceinte Sacrée. Interestingly, the length of cella 17 of Ishtar Temple is repeated threefold in the Enceinte Sacrée, with the same width. The width of the building is a discriminating factor. Four distinct widths can be identified: 3.70 m, at the Ishtarat Temple and chambre des banquettes of the Ninhursag Temple; 5 m in three cases: Innana Zaza, Ishtar Ush cella 18, and the Temple of the Lord of the Land level II; the width of two rooms is 7 m: room XLVI of the Enceinte Sacrée and cella 17 of Ishtar. The cella of Ninhursag is 8 m wide. Two buildings present an exceptional width of 10 m: the Shamash Temple and the Temple of the Lord of the Land level 3, with a square room, which was unusual at that time.

Most of these temples are usually considered bent-axis temples: usually, one or two passages on one of the long sides of the room provide access to the room, with a specific installation set along one of the shorter sides. Two doors provide access to cella 13 of the Inanna Zaza Temple; room XLVI of the Enceinte Sacrée presents two pairs of doors, one providing access to the almost square courtyard XXVI and the other to the southern corridor. These two doors present two pairs of pedestals, indicating a form of closure. Two doors provided access to the Temple of the Lord of the Land: one to the vestibule and the other to a dependency. In this case, the two altars were in front of the entrance, not on one side of the room. The overall pattern of circulation was of the bent-axis type from the monumental entrance facing west, then north, via a transition room (Figure S3). In the others cases, only one door led to the cella; in two cases, the door was located in the middle of the long wall; in the four others, the door was near one of the sides, to the left or the right of the room, as seen from the outside. The diversity of the organizations of these temples has long been underlined.

**Table 1.** Mari, ceremonial rooms' length and width, ordered by surface area.

| Room | Length, m | Width, m | Surface Area, m$^2$ | Rate Length/Width | Datation |
|---|---|---|---|---|---|
| Ninhursag chambre aux banquettes | 8 | 3.70 | 29.6 | 2.16 | Ville II ECM 4/5 |
| Ishtarat | 10.5 | 3.50 | 36.75 | 3 | Ville II ECM 4/5 |
| TSP level II | 9.8 | 5 | 49 | 1.96 | Ville II late ECM 5 |
| Cella 18, Ishtar Ush | 11.5 | 5 | 56.5 | 2.3 | Ville II late ECM 5 |
| Inanna-Zaza | 14 | 5 | 70 | 2.8 | Ville II ECM 4/5 |
| Cella 17, Ishtar Ush | 9.50 | 7.5 | 71.25 | 1.27 | Ville II Early/late ECM 4/5 |
| TSP level III | 9.8 | 9.8 | 96.04 | 1 | Ville II Early ECM 4 |
| Ninhursag cella | 12.90 | 8 | 103.2 | 1.61 | Ville II ECM 4/5 |
| Shamash | 15 | 9.8 | 147 | 1.53 | Ville II ECM 4/5 |
| Enceinte Sacrée, room XLVI | 29 | 7 | 203 | 4.14 | Ville II ECM 4/5 |

Apart from the classical typology distinguishing the salle oblongue and salle carrée established by Margueron (Margueron 2004, p. 250), a different order the buildings can be considered according to the width of their rooms, which are all oblong and, in most cases, with a bent-axis approach. We can distinguish four levels. As the width raises, the length does not necessarily. The ratio of the length to the width varies considerably, from one to three, with three steps: 1:1, for almost square temples (TSP III, Ishtar cella 17 and Ninhursag); 1:3, for Ishtarat, Inanna Zazam and Enceinte Sacrée; and in the middle, with an average of 1:2, for Ishtar *cella* 18, Ninhursag Chapel, and TSP II. the almost square temples present an area of 96.04, 76.25, and 63 m$^2$, respectively. The average area of those rooms is 77.31 m$^2$ and the three temples are characterized by a width between 7 and 9 m. Additional area is gained by elongation of the rooms, as is the case with Shamash Temple and the Enceinte Sacrée, at 147 and 154 m$^2$, respectively. At a lower level but with a width of 5 m, this is also the case for Ishtar cella 18 (56.5 m$^2$), Inanna Zaza (70 m$^2$) and TSP II (49 m$^2$). This is also the case for the smaller rooms at Ishtarat (36.75 m$^2$) and the chambre des banquettes of Ninhursag Temple (29.6 m$^2$).

Combining all these factors, I propose that at Mari, ville II, there existed four ranks; two low levels (ca. 30 and 55 m$^2$); a monumental standard with square temples, with an average surface area of 70 m$^2$; and two exceptional monuments, again rectangular, with an average area of 150 m$^2$. The distribution of these rooms in the monumental center is interesting to analyze: the monumental standard (Rank 2) presents two localizations: the center of the city with two adjoining major temples and Ishtar Ush cella 17 near the inner wall of the city. Rank 1 comprises one temple in the center of the city and the other is the Enceinte Sacrée. Rank 3 comprises *cella* 18 and TSP II, both the result of later developments in the history of city II, with Inanna Zaza, which might also have been a late development. Rank 4 comprises the smallest ones in the monumental center and, possibly in both cases, linked to a larger cella.

We know that the Temple of the Lord of the Land was the first temple built in Mari ca. 2500 BC (Butterlin 2021b), before the palace and its Enceinte Sacrée. It may be that the square formula is the oldest, before the adoption of the various rectangular temples with a bent axis. Combined with a stepped terrace, the Temple of the Lord of the Land constitutes the main sanctuary of the city. The Ninhursag Temple is a particular building: apart from its massive cella, with an enormous southern wall, its orientation is different from the other buildings of the monumental center, probably an inheritance of older installations (Beyer 2021a). This temple has also provided an exceptional number of votive deposits and favissae (Beyer and Marylou 2007; Obreja 2021), with a very specific set of objects, notably the famous eye stela (Beyer and Marylou 2007; Margueron 2007) and an eye idol. They are much older than the city and probably came to Mari with its founding people (Butterlin 2021d).

Notably, cella 17 of Ishtar is also almost square and presents a larger wall through which the door passes, to the north this time. This could be an indication of its seniority. I suggested elsewhere that this building was the place of coronation, linked perhaps to three monumental tombs situated beneath it, the possible burial place of three royal ancestors (Butterlin 2021b). The three temples also present specific installations (two low podiums in Ishtar and Ninhursag) and altars in the latest phase at the Temple of the Lord of the Land with a frontal approach and an off-center door. To summarize, these three buildings could have defined the roots of the city: the Temple of the Lord the Land, as major deity, with its high terrace; the Ninhursag Temple, embodying a long local tradition; and the Ishtar Temple as source of the power of the royal clan.

I suggest that the other temples are the result of the development of the city, both as a royal center (the Enceinte Sacrée of the palace and cella 18 at Ishtar Ush) and as attractive places for other actors in the city, with the temple situated beneath the central platform of the Ninhursag and the Lord of the Land (probably the God Dagan and his wife Shalash/Ninhursag). These temples, the temples of the city, present various areas, with Shamash being the biggest, followed by Inanna Zaza and Ishtarat, both of which are bent-axis temples. The Shamash Temple cella remains poorly known.

### 2.2. The Early Dynastic Temples of the Diyala

Since their discovery at the beginning of the 1930s, the temples of Mari have been compared to the temples excavated at the same time by the Oriental Institute of Chicago in the Diyala (Parrot 1956, 1974; Delougaz and Seton 1942). Both architecture and the inventory of the temples show obvious similarities, although it quickly appeared specificities existed, well-outlined by Tunca, for instance (Tunca 1984). The Dialya record provided a much longer sequence of development of temples (from the end of the fourth millennium to the Akkad period) than the Mari temples, contemporary with Early Dynastic III a and b in Central Mesopotamia (Figure S6).

I considered 28 rooms (Table 2), grouping together identical rooms reconstructed from level to level, as is the case at the Sin Temple, level I to V, or VI–VII. The width of the rooms ranges between 2.70/3 m and 5.50 m. They are divided into four classes:

- The width of the cella of the earliest Sin temples (level I to V), the single shrine at Asmar, and the shrine III of the square temple is 3 m. This is also the case for the small temples, level B–D. The width of *cella* 51 at the Nintu Temple is 2.70 m, 2.80 m for the small temples (levels A and E–F), and the small shrine, for 13 rooms in all.
- The width of the of the sin temple (level X), the archaic shrines at Asmar, and small temples H and I is 3.5 m, for a total of six examples.
- The width is 4 m at Khafadgé, Sin Temples VI to X, three rooms in the Nintu Temple, and another room in the square temple at Asmar, for a total of seven rooms.
- At last, two cellae are wider: the third cella of the square temple (4.8 m) and the cella of the Shara Temple at Agrab (5.5 m).

Table 2. Diyala Early Dynastic temples, ceremonial rooms, length, and width, ordered by surface.

| Room | Length | Width | Surface (m²) | Rate | Datation |
|---|---|---|---|---|---|
| Small temple A | 7.5 | 2.8 | 21 | 4.47 | ECM 2 ED I |
| Small temple F | 8.6 | 2.8 | 24.08 | 3.07 | ECM 4 |
| Small temple E | 9.2 | 2.8 | 25.76 | 3.28 | ECM 2 ED I |
| Square Temple, shrine III, E 17:20 | 8.7 | 3 | 26.1 | 2.9 | ECM 5 |
| Small temple D | 9.2 | 3 | 27.6 | 3.07 | ECM 2 ED I |
| Small temple B | 9.4 | 3 | 28.5 | 3.13 | ECM 2 ED I |
| Small temple G | 8.8 | 3.4 | 29.92 | 2.59 | ECM 4 |
| Archaic shrine I | 9 | 3.5 | 31.5 | 2.57 | ECM 2/3 |
| Archaic Shrine II | 9 | 3.5 | 31.5 | 2.57 | ECM 2/3 |
| Small temple H | 9 | 3.5 | 31.5 | 2.57 | ECM 4 |
| Square temple, shrine II D17:9 | 8 | 4 | 32 | 2 | ECM 4 |
| Small temple I | 9.2 | 3.5 | 32.2 | 2.63 | ECM 4 |
| Nintu 51 | 12 | 2.70 | 32.4 | 4.44 | ECM 3 (?)/4 ED II |
| Sin I–V | 11.70 | 3 | 35.1 | 3.9 | ECM I-II |
| Single shrine | 12 | 3 | 36 | 4 | ECM 5/6 |
| Nintu 4 | 10 | 4 | 40 | 2.5 | ECM 3 (?)/4 ED II |
| Sin VI–VII | 12 | 4 | 48 | 3 | ECM 2 |
| Nintu 52 | 12.5 | 4 | 50 | 3.12 | ECM 3 (?)/4 ED II |
| Square temple, shrine I D17:8 | 11.5 | 4.8 | 55.2 | 2.39 | ECM 2/3 ED I |
| Sin X | 17 | 3.50 | 59.5 | 4.86 | ECM 5 ED III b |
| Sin VIII–IX | 15 | 4 | 60 | 3.75 | ECM 4 ED III a |
| Sin X | 17 | 4 | 68 | 4.25 | ECM 5 ED III b |
| Agrab | 18.5 | 5.5 | 101.75 | 3.36 | ECM 5 ED III b |

The average length of these rooms ranges from 5.8 to 18 m, with a slight progression. Four shrines are between 8 and 9 m long (with a width of 3.50 to 4 m), 11 sanctuaries are between 11 and 12 m, one is 15 m long (Sin VIII), and four range between 17 and 18 m. The width of the 11 sanctuaries is 3 to 4 m, and the ratio of width to length varies from 1:3 to 1:4 at Sin Temple; 1:3 at Nintu Temple, the small temples, and the archaic Shrines; and 1:4 at the single shrine.

In the square temple, the three cellae present huge discrepancies in width, length, and in surface area. The ratio is less then 1:3 in the three rooms, especially in shrines I and II. These shrines seem to stand out from the rest, as does the specific layout of the three

shrines around a courtyard. Shrines II and III are smaller (32 and 26.1 m$^2$, respectively) compared to the 55.2 m$^2$ of shrine I, which stands out due to its dimensions and its double access: one from the courtyard with an alleyway in bitumen leading to it and the other leading to the priest room.

The average surface area of all but one of the shrines at Tell Asmar, the surface area of the first levels of the Sin Temple, and level B of the small temples is 30 m$^2$. The average area of these shrines is 48.86 m$^2$, reached at Sin Temple levels VI–VII, two of the shrines at the Nintu Temple, and of shrine I at Tell Asmar. This surface area is exceeded by later developments at the Sin Temple (from 60 to 68 m$^2$) and the Shara temple, which is an outlier with a surface area of 101.75 m$^2$.

A hierarchy is defined in four levels: the ca. 30 m$^2$ shrines, with a width between 3 and 4 m and a length between 8 and 11 m; the ca 50 m$^2$ sanctuaries, typical of ED II shrines, the later Sin Temples (from 60 to 68 m$^2$) and on top the Shara temple with 10,175 m$^2$. During ED III, we observe both an enlargement in the sanctuaries, especially at Sin Temple, which is merely an elongation of the rooms; and the Shara temple, appearing as another level in a much more monumental fashion.

This four-level hierarchy has two dimensions: chronological and functional. The former is linked to tripartite (Sin Temple I–V) or bipartite units (archaic shrines), and the latter either to bipartite (Sin VI–VII) or monocellular rooms (Nintu or square temple shrine I). The third level is either bipartite or, interestingly tripartite again, as is the functional unit around the Shara Temple cella.

Regarding the access system, a distinction must be drawn between tripartite systems and bipartite or single-room systems. As observed by Delougaz, the tripartite system gives way to an organization in which the central room is not the end point of a route but a through room with a specific focal point. Bipartite and single shrines are the end point, and the question is to know if this has religious significance. In the case of tripartite buildings, the Sin Temple presents an asymmetric layout of doors, four to the east and two toward the west. Two of the eastern doors lead to back pieces and two others lead to through rooms and to the courtyard, a peculiar layout for tripartite buildings. Two other doors lead to what is interpreted as a staircase, a type of staircase unknown before in tripartite buildings.

Bipartite buildings present, in most cases, two doors, attested by Sin level VII–VIII, the two archaic shrines. It is also present at shrine I at the square temple, Nintu 51 and 52, and at the single shrines of Tell Asmar. This is also the case at Shara Temple at Tell Agrab and Sin X. In most cases, the two doors lead to different areas: one is an entrance from the outside, from the courtyard, or a through room; the other, usually beside the main podium, lead to a dependency. In the case of shrine I in the square temple, it is the opposite: the main entrance is situated near the podium and at the end of a bitumen alley, probably a ceremonial path linking the cella to the ablution room. The other cases present only one door, usually situated in the corner opposite the main podium (Sin VII and VIII, Nintu A, and Shrine II and III at the square temple). There is no link between the number of accesses to the main room and their surface area. Apart from shrine I of the square temple, a general rule is the situation of the main access on the far side of the podium, the classical bent-axis approach.

### 2.3. Looking toward the South, Southern Mesopotamian Experiences

Apart from the North Temple and the Inanna Temple in Nippur (Table 3), it is well-known that the records for the temples of Early Dynastic Mesopotamia are scarce. There were clearly two different types of buildings: the bent-axis temples in Nippur and the axial buildings excavated in Adab and Girsu/Tello, which were different.

**Table 3.** Nippur, ceremonial rooms' length and width, ordered by surface (two lengths are provided when the room is not rectangular).

| Nippur/North Temple | Length m | Width m | Surface | Rate | Datation |
|---|---|---|---|---|---|
| IX | 9.20/8.40 | 3.5 | 30.8 | 2.74 | ED I ECM 2 |
| VIII | 9.20/8.40 | 3.5 | 30.8 | 2.74 | ED I ECM 2 |
| VII | 9.20/8.40 | 3.5 | 30.8 | 2.74 | ED II ECM 2/3 |
| VI | 9.20/8.40 | 3.5 | 30.8 | 2.74 | ED II ECM 2/3 |
| Inanna temple VIII–VII | 8.20 | 4 | 32.8 | 2.05 | ECM 4 ED III |
| V | 9.40/9.20 | 3.8 | 35.34 | 2.44 | ED II |
| V | 9.50/9.20 | 3.8 | 35.34 | 2.44 | ED II |
| III | 9.60 | 4.40/4 | 40.32 | 2.28 | ED III ECM 4 |
| IV 3–2 | 9.80 | 4.40/4 | 41.16 | 2.33 | ED III ECM 4 |
| IV 1 | 9.80 | 4.40/4 | 41.16 | 2.33 | ED III ECM 4 |
| II | 11 | 4-5 | 49.5 | 2.4 | Akkad |

The North Temple at Nippur (MacCown et al. 1978) presents an impressive set of buildings, with 10 levels that provided a succession of mostly trapezoïdal cellae, first in length and later, from level IC 3-2, in width (Figure S7). From ED I to ED III, the width grows regularly from 3.5 m from level IX to VI inclusive, 3.8 m on level V, 4.40/4 m from level III to II, and then 4–5 m on level II. This slow growth in length is matched largely in length from 9.20 to 9.80 m, with the last levels marking first a reduction (level III) and then a significant elongation at 11 m. The ratio of width to length is slowly reduced, from 1 to 2.74 to 2.28, which means that as the width increased, the length was proportionally growing more, a process observed in many cases. As for the surface area, the whole set of temples presents clearly three steps at 30, 35, and 40 m$^2$. Interestingly, North Temple level IX is 35 m$^2$ at the beginning before a reduction to 30 m$^2$.

This type of bent-axis cella or chapel is also demonstrated at the Inanna Temple level VIII, VII, locus 179. This room, 4 m wide and 8.20 long, is quite similar to the North Temple cella. Situated south to the main square cella, this room was clearly of special significance to with its benches and associated favissae (A and B). The main sanctuary was composed of one square cella and a vestibule, a completely different organization: the square room 178, 5 × 5 m, was accessed through an in-axis door.

This type of sanctuary has been compared, with caution, with other buildings excavated in Southern Mesopotamia (Figure S8): the Adab Temple (Wilson 2012, pp. 88–89; Marchesi and Marchettit 2021, pp. 47–49, who discuss Wilson's reconstruction and suggest a double cella) and the two buildings discovered at Ningirsu Temple (the *construction d'Ur Nina* and *construction inférieure* (Marchesi and Marchettit 2021, pp. 38–44)). In all these cases, the focal point of the sanctuary is a square room accessed directly or through a vestibule by an in-axis entrance, associated with a smaller room (a broad room at Girsu and a square room at Adab). The size is similar: 5 × 5 m in every case. These temples belong to a different category of building and are, in the case of Adab and Girsu, situated on top of a high terrace. They interest us because of their square layout, which is displayed at Mari on a different scale but not at Diyala.

### 2.4. Looking toward the North

A close link between the temples of Central Mesopotamia and some temples in Northern Mesopotamia has been long ago observed, first in Ashur, and in the temples excavated at Yorgan Tepe, Nuzi (Figure S9). The Ishtar Temple at Assur (Heinrich 1982, pp. 126–28, ab. 191–194; Bär 2003, pp. 37–38, Lawecka 2019, p. 154), levels G and F, was the first excavated temple with a bent-axis access, considered at that time by Andrae as Hurrian. It is usually dated to the ED III period, now ETG 6 (level H) and 7 (Level G) in the Arcane chronological frame (Renette 2018; Lawecka 2019). Since then, many temples of that kind have been discovered. Apart from the Nuzi temples, which are usually forgotten in the discussions, such buildings have been observed in Subartu land (including Tell Taya). They comprise the "seven shrines of Subartu" as baptized by Matthews (Matthews 2002) for the North Jezira of the early third millennium and of the ED J 3 periods; the bent-axis temples of the kingdom of Nagar (Figure S10) (Lebeau 2006, 2020, for the Jezira buildings; Novak 2015 for the Middle Euphrates buildings). Porter (Porter 2012, pp. 178–80, fig. 23, 179) added two more examples to the seven shrines. Adding the Ishtar Assur temple the Nuzi shrines, the Tell Taya building, and the Nagar kingdom buildings, the complete record includes 19 shrines (Table 4), most of which date to EJ 3 or 4 (ca. 2500–2300 BC).

**Table 4.** Northern Mesopotamia, ceremonial rooms' length and width, ordered by surface area.

| Cella | Length, m | Width, m | Surface Area (m$^2$) | Rate | Datation |
|---|---|---|---|---|---|
| Raqa'I | 5 | 4.5 | 22.5 | 1.1 | EJ 2 |
| Brak HS 4 | 8.5 | 4.5 | 38.25 | 1.88 | EJ 2 |
| Tell Taya | 9 | 5 | 40 | 1.8 | ETG 6-9 |
| Beydar D | 7.1 | 6 | 42.6 | 1.18 | EJ 3 b |
| Beydar B | 8.6 | 5.3 | 45.58 | 1.6 | EJ 3 b |
| Nuzi F | 11.6 | 4 | 46.4 | 2.9 | ETG 9 |
| Kashkakuk III | 8 | 6 | 48 | 1.3 | EJ 2 |
| Beydar C | 8.85 | 6.45 | 57.08 | 1.37 | EJ 3 b |
| Beydar, square temple | 8 | 8 | 64 | 1 | EJ 3 b |
| Beydar A | 9 | 7.5 | 67.5 | 1.2 | EJ 3 b |
| Brak, FS level 3, 1 | 9 | 7.7 | 69.3 | 1.17 | EJ 3 b/4 a |
| Nuzi F G 53 | 12.2 | 6 | 73.2 | 2.03 | ETG 9 |
| Halawa level 1 | 10 | 8 | 80 | 1.25 | EM 2 |
| Nuzi G | 13.3 | 6.5 | 86.45 | 2.04 | ETG 7-8 |
| Beydar, White hall | 9.8 | 9.4 | 92.12 | 1.04 | EJ 3 b |
| Brak SS level 5, 23 | 12.4 | 8.9 | 110.36 | 1.39 | EJ 3 b/4 a |
| Halawa level 2 | 12 | 10 | 120 | 1.2 | EM 2 |
| Ashur Ishtar G | 19 | 6 | 123 | 3.16 | ETG 6-7 |
| Mozan | 16.5 | 9 | 148.5 | 1.83 | EJ 3 |
| Beydar E | 20 | 16 | 320 | 1.25 | EJ 3 b |

The excavated temple closest to the Diyala and Mari Temples is the Ishtar Temple at Assur, level G 2. Its layout, its ritual installations (notably called barcasses, the same type as in the Mari temples), and its furniture offer a remarkable set for comparison. The cella of the temple is a rectangular room 6 m wide and 11.25 m long. Its presents a wide portal on the north side opening toward a rectangular room (designed as an adyton), almost completely occupied by a low platform, bordered by five barcasses, similar to the ones from Mari (Heinrich 1982, p. 127). Toward the south, a door leads to a dependence. This

type of organization with *adyton* partially matches the situation observed in the Enceinte Sacrée. With one access and a global surface are of 123 m$^2$ (including the *adyton* added), it is one of the biggest cella observed in Upper Mesopotamia.

At Nuzi, levels G and F (Starr 1939, p. 62, plan 6–7; Heinrich 1982, pp. 152–53, Lawecka 2019, pp. 152–53), two of the three temples present a comparable situation: a length of 12–13 m and a width of 6 m on average, while level F, a the reduction of the previous level, is a 4 × 11.6 m room, with the most substantial reduction occurring in width. Level G is dated to ETG 7–8 and level F to ETG 9 (Renette 2018, p. 32), contemporary with Akakd and Ur III periods in Southern Mesopotamia. They are among the most recent buildings of the corpus under scrutiny. This is also the case for the Tell Taya Temple erected during ETG 6 and used until ETG 9 (Renette 2018, p. 29; Lawecka 2019, pp. 153–54), with a length of 9 m and an estimated width of 5 m.

The Mozan Temple is a peculiar building, situated on top of a high terrace. With a width of 9 m, a length of 16.5 m (Pfälzner 2011, pp. 179–80), and a surface area of 148.5 m$^2$, it is by far the biggest cella discovered in the region. This is also the case with the Halawa building, level 2: with a width of 10 m and a length of 12 m, this building situated on top of a terrace is also unique, with an in-axis entrance and the platform situated to the right. It is also one of the earliest buildings of this kind in Northern Mesopotamia, replaced on level 1 by a broadroom temple with an axis entrance and an almost in antis layout, two pillars flanking the entrance on its left and right. It is smaller than the previous building (8 × 10 m). Porter considers the earlier level 3 buildings, 309, 312, and 313, as shrines, but this remains largely hypothetical, as no specific installation has been recovered. In this case, the link to the sacred is based on the paintings recovered, particularly in room 312.

Let us look at the Nagar kingdom temples in Tell Brak and Tell Beydar. At Tell Beydar, Temple A is a large building 25 × 30 m in size with a cella (room 6682) 7.5 × 9 m in size, and a surface area of 67.5 m$^2$. Temples B and C, situated to the south of Temple A along a narrow street, present a similar layout to a courtyard, giving access to a bent-axis cella, not completely rectangular: in the case of Temple B, cella 32771 is accessed from the west, 5.30 × 8.6 m (45.58 m$^2$), in size and cella 32747 (temple C) is 6.45 × 8.80 m (56.76 m$^2$) in size. Temple D is situated on the eastern side of the main street, with a different layout of the access system of the temples and the two back rooms of the cella. However, the basic features are present, notably the small courtyard providing access to *cella* 14226, 6 × 7.10 m in size (42.6 m$^2$). Temple E (Lebeau 2020, p. 278) is a much larger building than the other with an impressive central space, defined by Lebeau as "un espace central cérémoniel" (13471), 20 × 16 m in dimension (320 m$^2$). This hall is considered a cella because the typical installations considered the benchmark of temples at Beydar were identified in the hall: a niched façade, with a bench at its foot and a podium to the north.

With such dimensions, it raises the question of its coverture (Pfälzner 2011, p. 182). Lebeau proposed fairly convincing arguments in favor of a covered hall, namely the fragile nature of its installations and the reinforcement of the eastern and western walls (Lebeau 2020, p. 280). The attribution of those temples to peculiar deities was discussed by Lebeau (Lebeau), with a list of divinities mentioned in the Beydar texts. It remains largely tentative, although the discovery of some objects in Temple C might be in favor of an attribution as temples. Among those buildings, the "white hall" stands out: situated to the east of Temple E and the south of the great square, this monocellular building (room 19333) presents two entrances: a low podium and a bench. It is considered a ceremonial building not specifically a temple, although Lebeau considered that a possible ritual function cannot to be completely excluded (Lebeau 2020, p. 277). The room is almost square, 9.40 m by 9.80 m. Since the main entrance is the northern door providing access to area S, the second access appears to be a secondary access to a corridor leading toward the temple E, which can be considered as a bent-axis room. These ceremonial buildings have been compared to the Brak temples, and the similarities are striking (Pfälzner 2011, pp. 182–83; Lebeau 2020).

Temple E was compared to the temple of Shamagan (FS complex, level 5) and the east courtyard of the SS complex, level 5, at Tell Brak. The dimensions of their central spaces

are difficult to assess because their layout is not completely known. However, the width of those rooms is known: 11 m at FS and 16 m at SS. In the latter case, it is a trapezoidal room. The excavators thought that the rooms were not covered but considering the analogy with Temple E excavated at Beydar, this may be the case. Two other "temples" were compared to the Beydar temples, namely, room 23 of the SS complex and room 1 of the FS complex, level 3. Room 23 is polygonal, with one door, a length of 12.4 m and two widths: 8.6 and 9.2 m (surface area of 110.36 m$^2$), to the east and west, respectively. Room 1 presents three doors and no specific installation. It is 7.7 × 9 m in size, with a surface area of 69.3 m$^2$. Strangely, Lebeau compared them to the almost square white hall, not to the other rooms considered as cellae at Beydar. Another square building was unearthed at Beydar, the square temple in Area F, slightly later than the other buildings (EJZ 4 c). At 8 × 8 m in size, it is smaller than the white hall, but is another example of this type of building with a different layout (central altar and one access).

All these buildings date from EJ 3 or 4, so the ED III and Akkad period. At Brak, Matthews discovered an earlier bent-axis cella at HS 4: with a width of 4.5 m and a length of 8.5 m, it is the earliest example in the Nagar kingdom, dating from EJ 2. Two more bent-axis shrines have been discovered along the Khabur at Raqa'I and Kashkakuk III (Pfälzner 2011, pp. 177–79). The temple discovered at Raqa'I, level 3, is the smallest building of this sequence, 4.5 × 5 m in size (Schwartz 2000, fig. 5). As at Kashkakuk, the temple excavated in area A, level A IV, is 6 × 8 m in size.

Apart from the Raqa'I building, this set of cellae is fairly homogeneous. An average 6 m width is quite common: 7 of 17 buildings present this width, particularly at Tell Beydar, but not at Tell Brak, where the width more commonly ranges between 7.7 and 8.90 m. This width is attested in two cases at Tell Beydar, temple A and EJZ 4 square temple. The next step of this hierarchy is 9 m, with the white hall, room 23 at FS and the Mozan temple; Halawa Temple, with a width of 10 m, is unique in its layout. This coverage scope is found three times. To exceed that scope, the question remains of the mega halls at Brak and Beydar with a scope of 16 m, which is the maximum found during the third millennium. It is technically possible but it is obviously a specific layout and scale. With the less-common 4 m width (only three cases), we can define a five-tier hierarchy of cellae with an average width of 6.60 m (if Temple E is not considered), up to 7.14 m.

Considering the length of the buildings, we can define three clear clusters. Apart from the earlier Raqa'I temple, most buildings have a length between 8 and 9 m (seven cases). At Brak and Nuzi, the length of 12 m is the second level; beyond, we find the great halls, with a length between 16 (Ashur and Mozan) and 20 m (the great halls of Beydar Temple E and Brak). A three-tier hierarchy appears. Considering the ratio between length and width, a general tendency appears, already noted by Lebeau, toward square or more or less square temples. The ratio between length and width is, on average, 1.5 in the Jezira cellae, and the width increases with the length. This ratio remains the rule even with the greater temples, such as at Mozan or Temple E. It is not the case in the Tigris region, where 1:2 or 1:3 is the common rule. The average surface area of those rooms is 62.72 m$^2$ if we exclude the great halls, and 92.77 m$^2$ if they are included. Most of the cellae fall into two groups: those between 40 and 50 m$^2$, with four cases; those between 60 and 75 m$^2$, with five cases. Again, the Raqa'I cella is at the lower margin, and beyond 75 m$^2$, we can identify two groups: the great halls with surface areas above 270 m$^2$ and the great temples at Mozan and the Ishtar G Temple at Assur. Apart from the great halls and Brak SS room 23, all the Jezira cellae are much smaller and are grouped in the medium groups, especially at Beydar. One may wonder if the difference between these standardized buildings has a meaning.

This type of cellae is demonstrated from EJ 2 and most common during the EJZ period. They are integrated at Tell Brak and Tell Beydar in monumental complexes that are multifunctional. They are therefore parts of a monumental grammar that combines the different kinds of structural units well-identified by Lebeau: great courtyards, bent-axis temples, and great halls. The different types of buildings is, as in Mari, a puzzling

question: if these buildings are temples, does it mean that different kind of rituals occurred or different deities were worshipped?

I studied 70 rooms considered, usually the cellae of bent-axis temples, situated in Southern, Central, and Northern Mesopotamia. Their distribution is uneven in time and space (Figure S10). I identified specific characteristics in each region, which are included in this review. It is time now to discuss the results considering the whole set.

## 3. Discussion

In order to discuss the results, we have to consider different levels. First, this type of bent-axis temple already appeared at the very beginning of the third millennium in Central Mesopotamia, in Diyala and at Nippur, and slightly later during EJZ 2 in Upper Mesopotamia (Figure S11). Two cases not discussed here are also attested in the Southern Levant at Jericho and Ell Tell (Manfred 2003, pp. 24–25) the other temples being of the broadroom type. Bent axis temples are the dominant formula in Central Mesopotamia and Jezira during the EJZ 3 and ED III periods, notably at Mari, where they appeared probably just after the foundation of City II, ca 2500 B.C., not at its beginning. It is not the only layout used in temples at that time: in the north, notably the Euphrates but also in the Jezira, there appears "in antis temples", which tend to replace the bent-axis temples, particularly along the Euphrates (EME 3 and 4 in the northern Middle Euphrates), in Jezira (EJ 3 a at tell Khuera), and the Shakkanakku period in Mari (from 2200 B.C. onwards). In Southern Mesopotamia, square in axis temples are mostly identified in ED III, but the documentation remains limited. Moreover, this type of temple disappears almost completely after the Akkad period (except in Ashur and Nuzi), during which some of these temples remained in activity, and one can also question the reasons for that process. To summarize, this specific layout seems to be more or less contemporary with the development of city states in Central and Upper Mesopotamia from the Tigris to the Euphrates Rivers. The link between these developments remains to be established.

### 3.1. Bent-Axis Temples and Ancestor Cult

Since the 1930s, this type of bent-axis approach has been considered an inheritance from the fourth-millennium Uruk temples, notably the white temple in Uruk or the Uqair Temple. This assumption is largely based on the Sin Temple sequence in Khafadgé as analyzed by Delougaz and Lloyd (Delougaz and Seton 1942). They noticed that the progressive evolution toward bipartite or single shrines transformed the conception of space: not a passage through building but a dead end, with a face-to-face between the worshipper and the divine, usually mediated through votive material. Evans criticized this view of the sacred space built upon the theories of Andrae (Andrae 1922, table 11a) or Lloyd (Delougaz and Seton 1942, fig. 159), with a scenography almost thought of as a kind of museum setting, with statues aligned on benches along the walls (Evans 2012, pp. 78–81). This approach was founded on the idea that a specific relation to the sacred was linked to a rotation of 90° to the right or to the left, to face the divine podium, which was not visible from the outside as in an in-axis temple, for example.

That this layout conveys a different conception of the sacred remains a debated issue, but it is a material fact, as how the cellae were inserted in different environments, notably tripartite, bipartite, or single structural units, and were usually associated to courtyards and dependencies. They are part of a religiosity of movement. The cellae are the culmination of religious itineraries, not dead ends. These routes are punctuated by ritual installations, which lead from the outside to the inside and vice versa.

It has long been considered that tripartite units disappeared after ED I, but the discoveries in Beydar show a revival of the formula, though along completely different rules. We observed the elongation of the Sin Temple cellae, in complete contrast to the almost squarish cellae at Beydar. Trying to understand these rooms as an expression of religion has been the ambition of phenomenologists, notably Tunca (1984) and Margueron. Combining the traditional typological analysis initiated by Andrae and Heinrich, they have

studied ritual installations, and Margueron proposed a tripartite approach of the sacred (Margueron 1995). This fundamental form is largely based on Parrot's views of the Mari temples as organized in the same way as the Jerusalem Temple, as we have seen. Religious activities were not confined to the cella but were also occurring in forecourts.

This especially means that our approach to this architecture is just a first step toward a new way of assessing the data. On different scales, Evans and Ristvet (Evans 2012, chp. 4; Ristvet 2013 showed that the assessment of these buildings has to been integrated in a dynamic approach, both in time and space. They are part of an urban fabric, regularly regenerated through complex ceremonies occurring at the level of the city or the building. Regular reconstructions and resacralization processes are cyclical and involve the architectural setting, the ritual installations, and the life and death of objects. In this context, the interpretation of bent-axis temples has slowly shifted from a traditional approach to sanctuaries to other types of discussions. The first step is the proposal that the statues discovered in Northern Mesopotamia at Khuera were part of an ancestor cult, and since then, the question has regularly been reopened, for example, by Ann Porter about the Subartu Shrines (Porter 2012) or Tunca about the protodynastic statues (Tunca 2016). Additionally, Forest (1996) and Lawecka (2011) discussed the actual function of the Diyala sanctuaries as temples.

It is well-known that the identification of the temples is founded upon inscribed votive objects, dedicated to a deity, as is the case for Diyala and Mari, although in the latter case, some identifications relied upon later buildings. This is the case for the Ninhursag building, identified as such by foundation deposits from the Ville III at Mari. The Ville II temple was considered a precursor, even if Ninhursag does not appear as such in the offering lists (Lecompte 2021). This is typically a case of retrograde movement of reality, founded on the idea of continuity of the sacred place. In the other cases, the inscription states that a character has dedicated a statue or vase to a god. Since the same god is mentioned on statues discovered in one specific building, that was enough to identify a "temple", that is, the house of a god, even if the word does not appear itself. For example, among the fourteen fragments of statues discovered in 2009 in the favissae of the Temple of the Lord of the Land (LUGAL DINGIR KALAM), four mention this deity, probably the god Dagan (Figure S12). If it is obvious that the space in question was placed under the patronage or protection of a deity, how that deity was present and honored remains to be defined precisely. No divine statue has ever been recovered, for instance, in Mari among the 600 statues; it is therefore preposterous to analyze those spaces using much later data. Religion built itself slowly in ancient Mesopotamia, as the cities and possibly processes were deeply entangled.

Forest (1996) argued that the bent-axis rooms may have derived from secular architecture from the Uruk period, especially the bipartite and monocellular reception halls observed at Jebel Aruda and Habuba Kabira, the famous colonial settlements built by the Urukeans on the Middle Euphrates during the fourth millennium. Additionally, Porter has proposed that those buildings or units were devoted to the ancestor cult (Porter 2012, pp. 189–94). Both for bipartite units and single shrines, which developed at the beginning of the third millennium in Upper Mesopotamia, she argued that they played a ceremonial role, directly linked to the actual scale of the rooms. She opposed the vast halls of the Uruk period, who, in her view, could welcome more than 300 people to these halls, which remain modest in size and could welcome only a handful of people. The actual scale of those buildings helps to define to which degree they were inclusionary or exclusionary for the people attending whatever rituals occurred there. At least it provides an idea of a maximum accommodation capacity (for people or objects). In Forest's view (Forest 1996), those temples were merely profane buildings used as community centers in urban neighborhoods; the size of the rooms may be an indicator of the social units linked to those buildings, whatever their function.

*3.2. From Proto-Urban to Early Dynastic Temples*

It is therefore all the more interesting to look at the results of my quantitative approach. It is but a first step toward the understanding of those social units and the type of scenography that was implemented. If we return to our 70 shrines (Figures S13 and S14), it is interesting to globally examine the whole record. Obviously, we are considering a rather heterogeneous set: the smallest room is 16.1 m$^2$ and the largest are the great halls in Brak and Beydar at 270 and 320 m$^2$, respectively), and the Enceinte Sacrée in the Palace at Mari. The average size of those rooms is 52.92 m$^2$. Many (28) of those shrines are rather small, ranging between 25 and 35 m$^2$, with most of them are situated at the Diyala sites, and dated especially from the earlier phases, that is, ED I or II. Two sizes are prominent, around 30 and 35 m$^2$.

Additionally, I suggest that there are four significant sizes: around 40 m$^2$, around 50 m$^2$, between 60 and 70 m$^2$, and from 86 m$^2$ onward. Around 40 m$^2$, two groups are paramount: the later north temples in Nippur and the Beydar temples (Temples B and D). Around 50 m$^2$, we find Sin Temples VI and VII, the last phase of the Temple of the Lord in Mari, and the latest Nippur North Temple. Between 60 and 70 m$^2$, we have 11 shrines, among them the Nuzi temples, Beydar temples (Temples A, C, and square temple), the bulk of the Mari temples (Ninhursag, Inanna Zaza, and cella 17 of the Ishtar temple), and the Brak temples and the later Sin Temples (VIII–X) at Khafadgé. There is a clear gap between those shrines and the next step, from 86 m$^2$ onward: four shrines range between 86 and 100 m$^2$, and it is a rather heterogeneous group of shrines with two square rooms (white hall at Beydar and TSP level III), the Shara temple at Agrab and the Nuzi G building, with two clusters around 110–120 m$^2$; Bark SS level 5, room 23 and the G level of the Ishtar Temple at Assur; and two again around 150 m$^2$, namely the Shamash temple in Mari and the Mozan Temple. As stated before, we find the Enceinte Sacrée and the great halls of the Nagar Kingdom.

It is interesting to compare those dimensions, as Porter suggested, to the previous proto-urban tripartite or bipartite building central spaces. I presented elsewhere the results of a study of those proto-urban central spaces (Butterlin 2018), and present here some insights and reflections about the possible link in terms of accommodation capacity or layout. In Uruk monumental architecture, I identified a monumental standard at ca. 90 m$^2$ both for the monuments at Eanna and the temples of the Anu Ziggurat or the Uqair Temple (Butterlin 2018, pp. 364–71). The buildings of the second rank are much bigger with central spaces that are around 300 m$^2$ in size; the biggest one is the Kalkteingebäude at 743 m$^2$. Rundpeilerhalle is 300 m$^2$ and Pfeilerhalle in Uruk is 200 m$^2$ (Butterlin 2018, pp. 378–80). If we try (with necessary caution) to compare those figures with our 70 shrines, it is obvious that the bigger sanctuaries of Early Dynastic Mesopotamia, that is, the Enceinte Sacrée and the Great Halls of Tell Beydar and Brak, match the second-rank buildings in Uruk in size, but not the giant buildings. The standard size of the reception room in monumental buildings at Eanna and Anu Ziggurat is 90 m$^2$ (Figure S15), with a width of 5 to 6 m. As I show below, this scale is comparable to the Agrab Temple (significantly a tripartite structural unit) and to the almost square buildings in Mari or Beydar, which present clearly a different layout.

This means that if size is considered, most of the shrines of the Early Dynastic are much smaller than the Uruk examples; they are more comparable to the surface of the kopfbau of the Uruk buildings, usually 3.5 to 4 m in width. They can also be compared to the central spaces of two peculiar and enigmatic buildings, the Riemchengebäude, with a central space of 3.50 m × 6.16 m' and the Steingebäude, 5.27 × 10.35 m. In both cases, the length to width ratio is 2:1 (Ricardo 2007; Butterlin 2018, pp. 345–67), which is well-identified in the shrines of the first half of the third millennium in 15 cases, mostly at Nippur North Temple in two secondary rooms in Mari and Nuzi. In the case of the Uruk buildings, the Riemchengeäude is the first occurrence of a layout that could be interpreted as bent axis, although no specific installations were discovered in the central space. Steingebäude is the first occurrence of a broadroom in-axis plan. Both buildings have been interpreted as ceremonial buildings: the

Steingebäude as cenotaph and the Riemchengebäude as a building used to close a nearby temple or again as a building linked to the death of a king. Forest considered them as the only real temples in Uruk (Forest 1996), and they remain unique cases, which may have been a milestone in the evolution of sacred architecture.

The bent-axis temples are also much more comparable in size to the reception rooms of the 22 late Uruk houses I studied in the Middle Euphrates region, notably at Habuba and Jebel Aruda (Butterlin 2018, pp. 436–44). I proposed to distinguish among those houses a five-tier hierarchy, with the following steps: 15, 30, 40, 50, and 60–70 m$^2$. The temples of the Middle Euphrates present clear differences: at Jebal Aruda, they are around 40 m$^2$; at Habuba, they are 60 and 94 m$^2$. Those are the figures for the great houses, with tripartite buildings along the courtyard and sometimes additional bipartite or single units of reception. Among the bipartite examples, the three bipartite buildings excavated at Arslantepe are interesting: their reception rooms are 12 m wide and 5.40, 5, and 5.20 m long for Temple A, B, and building 37, respectively, with an average surface area of 60 m$^2$. As observed by Forest, those reception halls have much in common in scale with the Early Dynastic temples, and more with the Diyala and Nippur examples than with Mari or the Subartu Temples.

Here, I compare very different buildings: since it is usually considered that the former derive from the latter, there is an obvious conclusion. Whereas most of the 70 shrines range clearly in different ranks of proto-urban reception rooms of the great houses of the late fourth millennium, almost none can be compared to the monumental standard at Uruk, which is seldom attested after the fourth millennium BC. To the contrary, the upper level of the monumental prestige houses of Uruk is clearly present in a palatial context in Mari, and probably also in Brak and Beydar: are we dealing here a kind of great aulae? In Uruk, I proposed defining this upper level as "proto-palatial". Apart from the square temples of Mari and Beydar, all the other temples in Mari and Subartu rank between 40 and 70 m$^2$, that is, the biggest proto-urban houses but not temples. The typical Urukean monumental reception room (at the White Temple for instance) has virtually disappeared, a major break, I think, in the evolution of religious architecture.

On top of high terraces, at least in Southern Mesopotamia, we noticed that the typical size of the cella during the Early Dynastic is 25 m$^2$, more than one-third of the previous Uruk buildings. I elsewhere showed that this reduction occurred in the case of temples and with high terraces: at Uruk or Uqair the high terrace is 1500 m$^2$ in size and the high terraces of the Early Dynastic period are ca. 800 m$^2$ in size (Butterlin 2019). This is true for Southern Mesopotamia and at Mari (Figure S16). However, in the case of Mari, a low temple flanks the Massif Rouge toward the south, a square temple at level III 100 m$^2$, which is almost unique in layout. In Tell Mozan, a different and much bigger building stands upon the terrace.

In order to assess the scenography at work in those buildings, size matters, but the general layout and proportions of the room are of special significance. Width is of special significance, first because of the technical skills and the span of the wooden beams. Looking at the data, 45 of 70 the examples are less than 4.5 m in width, the minimum being 2.80 m. Most of our examples are situated in Diyala or Nippur. This width is typical of the proto-urban houses or prestige houses during the Ubaid and most of LC 1 to 3 in upper Mesopotamia. It is usually considered the result of the shortage of good-quality beams, notably in Southern/Central Mesopotamia, where the use of palm tree or poplar beams constrained the builders. Then, 5 m is a clear step, appearing in Southern Mesopotamia in LC 1 at Uruk, and I showed that the typical width of the standard monumental buildings reception rooms at Uruk is between 5 and 6 m. This is also the width of the temples of Arslantepe (Butterlin 2018, pp. 448–49). The major buildings such as Temple C, the Hallenbau for example, present central rooms with a width of over 8 m, with the top level being the Kalsteingenbäude and Temple D at 11 and 12 m, respectively. A width of 12 m is observed at the end of the fourth millennium, the maximum size. To cover these buildings,

imported pine or cedar wood beams were usually used, and they are found at Uruk, for instance.

The width of 5 to 6 m of 11 cellae of our group in shrines are situated either in Mari, or Upper Mesopotamia, notably Assur, Beydar, or Nuzi. It is also the dimension of the square temples of Southern Mesopotamia. This width necessitated good beams, perhaps more available in Northern Mesopotamia but necessarily imported in Mari and Southern Mesopotamia. This could explain, in part, why those temples are bigger than their neighbors of Central Mesopotamia. Twelve cellae present widths of 8 to 12 m, which corresponds to the top levels at LC 5 Uruk, and they are all situated in Mari or Upper Mesopotamia. The maximum width covered during the third millennium is 16 m, as already demonstrated by Margueron. This is the width of the great halls of Brak and Beydar, and the size of the almost square central room of the Enceinte Sacrée at Mari. Margueron (2004, pp. 219–20, fig. 207, p. 220) proposed that it was covered (opposed to Parrot, who considered it a courtyard), and it is interesting to find again the 16 m limit. This is a new stage in the race for gigantism, before the developments of the late third millennium in temple, terraces, and palatial architecture.

All these considerations are limited to technical constraints or the availability of local or imported material, and it is difficult to assess how far they were embedded in religious or symbolic thought.

### 3.3. Accommodation Capacity and Patterns of Circulation, Some Insights on a Religiosity of Movement

I think that one way to discuss this link is to evaluate the accommodation capacity of these buildings. In the case of bent-axis buildings, the situation is not similar to the fourth millennium buildings, especially the Urukean ones. The first obvious difference is the access system. The monumental Uruk buildings present a large number of doors, especially in Eanna, supposedly linked to the necessity of gathering and managing the flux of crowds, as in a theater. This is not the case at all with the bent-axis temples, usually accessed through one or two doors. Second, it is impossible to ignore the installations (ritual or not) present in these rooms. They constrain traffic and the gaze of the attendants or visitors, and the global capacity of the rooms. There are three main features that considerably limit the accommodation capacity: fireplaces or hearths, altars on one side of the rooms, and benches. In Uruk, it is well-known that the spaces presented either platforms or fireplaces (with the famous pan form), situated along the long axis of the rooms, centrally placed or not. Their layout by pairs or in one part of the room are good indications of how potential visitors were placed and I presented elsewhere some proposals about that disposal around the main fireplaces, possibly in circles (Butterlin 2021d, p. 40). The concentration in one place, as in Uruk Eanna, of proto-urban complexes arranged with the same modular units but in different ways suggests that different social units (impossible to identify as such) gathered there, under the protection of the goddess Inanna, patron of a proto-urban league of cities, Uruk being the center of a confederation. The typical layout of Urukean buildings (both monumental and domestic) disappears completely at the end of the fourth millennium after a major disruption, marked by the collapse of the Uruk system and the major restructuring of the Eanna center (from level IV to III).

The "hearth house temples" appear after that disruption; the first Sin Temples, albeit tripartite, are different from the previous Urukean ones, especially in the use of one aisle of the building for a longitudinal staircase, never found before. Hearths remain present but organized differently. This typical layout of proto-urban buildings, also present in earlier cases at Gawra, for instance, or later in Arslantepe, still exists in Diyala and at Nippur, albeit not systematically. From level III on at the Sin Temple, a small circular hearth is situated in the middle of the cella, as in archaic shrines I and II and in the small Temples C and G. At Nippur, from level IX to III, circular hearths are situated either right in the middle of the room or, from level V on, in the middle of the room opposite the altar. In Ashur, the Temple of Ishtar, a rectangular mudbrick construction (1.44 × 1.55 m) was also present in the middle of the room, interpreted by Andrae as a place of immolation, but

this remains doubtful (Heinrich 1982, p. 127). This is not the case in Mari, Beydar, or Brak, apart from the earliest HS 4 example. In these cases, the hearth or fireplace occupies the central part of the room opposite the platform.

In the case of the bent-axis temples, the small side opposite the entrance usually presents a platform, a podium either low or high, usually considered the place of majesty or epiphany. It occupies a significant part of the room and, in some cases, such as the Ninhursag temple in Mari, the low platform occupies almost one-third of the whole space.

The third recurring feature is the presence of benches. They are not systematically attested (notably in the Diyala), and are not necessarily running along all the walls. Since Andrae, it is considered that most of them are too narrow to welcome people and it is therefore considered that they were used to display statues. This point has been discussed. In some instances, notably in Mari (Beyer 2021b), the recovery of the foots of statues still, in situ, on top of some benches, and the presence of barcasses (elongated depressions in plaster or vessels embedded in plaster) indicated a place of libation or offering (probably to statues). This does not mean that all these benches that run sometimes along the walls were not used differently to welcome people.

If we consider that the ordering of the visitors in these rooms followed the benches, the actual capacity of those rooms may be assessed. We do not know the figures used in Ancient Mesopotamia, so there has been much debate about how to evaluate the accommodation capacity of these ceremonial rooms, either in temples or palaces. Both 1 person/m$^2$ and 2.5 persons/m$^2$ were proposed for evaluating the accommodation capacity of throne rooms in neo-Assyrian palaces (David 2019, p. 48), but in this case, a compact crowd of standing people was considered. In our case, this is not the solution considering the constraints noted above.

Given the lack of better options, we can apply the figures used today for assessing the capacity of a room used for a meeting (using a standard calculator of capacity of meeting rooms, https://fr.hotelplanner.com/Common/Popups/SpaceCalculator.htm, accessed on 23 July 2021). The solution is to use a U configuration (as in a conference room), which corresponds best to the situation we are considering. It eliminates one side of the room, in our case, the place of the altar, liberating the center and supposing a peripheral arrangement of people, seated, for instance, upon the benches.

For the first category of shrines, ca. 30 m$^2$, the resulting figure is 9; for 40 m$^2$, 12, and for 50 m$^2$, 15 people could be accommodated. For 70 m$^2$, the figure is 22, and for 100 m$^2$, 31 people. As for the greatest buildings, 200, 270, and 320 m$^2$, the capacities are 62, 83, and 99, respectively. In the last two cases (the great halls), it is probable that the actual configuration was different since they were devoid of benches. They could, at maximum, welcome 300 standing people. This figure is only a gross indication. Among the 70 shrines, benches are well-attested in Mari, Ashur, and Nippur, but seldom in Diyala. At Beydar, they are limited, situated besides the niched platform. This could also indicate a different configuration.

Even when benches are indicated, they seldom run along all the walls, and are interrupted by specific installations, as is the case in the Ishtar Temple at Ashur, or the Inanna Zaza Temple in Mari, which both present particularly well-preserved facilities. I provide here a tentative simulation of the situation in the Inanna Zaza Temple, if people were seated upon the benches. The surface of the room is 70 m$^2$, it could, following our calculation, welcome more than 20 people, but the benches are indicated only on two sides of the room: the northwestern side and the north eastern side. Twelve people (Figure S17) could actually be seated there, facing either the altar (only three persons) or the double entrances (nine persons). However, no bench runs along the opposite wall. The Inanna Zaza Temple belongs to the type of temples that were elongated, with a length-to-width ratio of 2.8. In this case, the elongation of the temple offers the possibility to welcome more people, or objects, with a width limited by technical constraints. At this point, the ratio of length to width is particularly significant, since it creates a completely different scenography.

As observed by Porter (2012, pp. 178–82), these figures provide an indication of the social units possibly involved in the use of those rooms, either virtually (by displaying statues) or physically. The 30–50 m$^2$ size could not welcome many people and could be linked to families or kin groups. Above, I dealt with different units, and the ca. 60–70 m$^2$ shrines, which are mostly attested in Upper Mesopotamia and Mari, could welcome theoretically more than 22 to 30 people. This provides an indication of a different social representation and significance of these "temples". Of course, we have no indication about the number of people actually allowed to enter the buildings or, specifically, the cellae, usually considered to have strictly limited access.

Another method of assessing the data is to consider the circulation pattern, not only a static pattern. Among the 70 shrines considered here, most of them (49) present only one entrance door from the outside either a courtyard or a transition room; 21 present two doors, usually placed at each end, apart at the Sin Temple in Khafadgé, with two doors located side-by-side. It is difficult to know if this specific arrangement of doors is linked to a circulation pattern. For example, one door could have been used for the entrance and the second for the exit. In two cases at Mari, at Inanna Zaza and the Enceinte Sacrée, the two doors leading to the sacred place are linked to alleyways composed of bitumen situated along three sides of the courtyard, but not the side abutting the cella. One could imagine a circumambulation scheme (Figure S18), with one-way traffic along the alleys on one side, then entering the cella from the entrance door (opposite the main place of worship) and exiting from the second door to the next alleyway and outside. In the case of a single door to the room, enter and exit would occur through the same way.

As indicated by the expressed bent-axis approach, the aim is that, to obtain a full view of the sacred place, the visitor has to turn 90°. In this case, the degree of visibility of the sacred place is dictated by the angle of clear vision, usually in between 60° for a full view and 90° for an extended one. Even for the square or squarish temples, as in Beydar, one cannot obtain a view of the altar or podium directly from the outside. The more elongated the cellae, the more this tendency grows. To return to our example of Inanna Zaza, with a ration of 2.8, upon entering from the northern door, the visitor would have had a full view of the opposite bench but not of the platforms, and plausibly of the whole set of people or objects displayed on the benches. They would have obtained a full view by turning 45° toward the right at least and later 90°. Interestingly, in the case of the north temple at Nippur, the part of the cella centered upon the hearth is visible straight from the outside, the possibly more sacred part (with the podium or altar) being visible only later. When the ratio of length to width increases, so does this degree of relative invisibility.

Assessing the global volume of these rooms remains difficult. A large majority of these shrines present a width inferior to 8 m and could therefore have had a second floor without experiencing technical difficulties. This possibility has seldom been considered and even in the rare cases where stairs are attested, they are usually considered to lead to a terrace. The idea that the most sacred place could be covered by a second floor is counterintuitive but has to be considered; its needs further study including the whole building. The question of the vertical development of these buildings is especially important with the almost square or square cellae and with the greater cellae of the corpus. The width of the walls is of special significance, with peculiar cases, as the Ninhursag temple or cella 17 in Mari: how can we explain the exceptional width of the walls crossed by the communication door?

*3.4. Does Size Matter? An Anthropology of Religious Spaces*

If these rooms were linked to a cult of the ancestors or worship under the patronage of a divinity, they could provide an indication of a different social system or, at least, on a different scale than the accommodation system. A working hypothesis is that we are dealing with a family-based system in Diyala and perhaps broader social units in Mari or the Nagar kingdoms, such as tribes, or ethnic or geographic entities, comprising those states that were, as in the case of Ebla, much larger states than in Central or Southern

Mesopotamia. The tribal component of the Ebla state is well-known, but we know almost nothing about the functioning of the Nagar or the Mari state at that time.

This consideration is just preliminary. In order to further enhance the study, we have to consider the role of the courtyards and central spaces of most of these. Firstly, the degree of isolation of the cellae first and, secondly, their actual urban environment are of utmost importance to understand how the urban fabric was conceived. The sanctuaries of Diyala and the north temple in Nippur are situated in a neighborhood environed by houses and supposedly linked to these houses (Figure S19). In Mari and Tell Beydar, most of the temples are concentrated in the upper center of the city; in Beydar, the temples are nearby the central palace (Figure S20); and in Mari, they are below the Massif Rouge (Figure S21). In all cases, the temples are clustered around a main road (Figure S22). In Mari, it has been named the *via sacra*, and in Beydar, the main street. The level of these cities is not the same: Nippur, Ashur, Mozan, and Nagar or Mari are considered as capital cities, possibly also Asmar. Beydar or Khafadgé were second-level cities subordinated to greater centers, as attested in the case of Beydar by the textual documentation.

If all these temples in Beydar or Mari were linked to a tribal or ethnic organization, their concentration was the actualization of a unitary process through a specific urban fabric. In Mari, the different hypostases of Ishtar/Inanna may well be linked to a geographic or ethnic affiliation. The Ishtarat is Inanna Sarbat, Inanna from the Bishri, and linked to the mountains dominating the steppe west to the Euphrates, a land of semi-nomadic people. "Inanna Zaza" is "Inanna of the poplars", which could be linked to the valley of the Euphrates (Lecompte 2021; Butterlin 2021c). In a capital city such as Mari, the concentration of temples in the monumental center can be considered the actual expression of social units that were not limited to the city, but to a broader territory, along the Middle Euphrates, the steppe land, and on the left and right of the Euphrates. The display of statues in theses sanctuaries, whatever their actual function (votive deposit by living people or representation of dead people) is of special significance, with the temple being the communal house of a specific part of the city-state, not only the city. In this case, the great halls, or the mega temples such as the Enceinte Sacrée, are the expression of an additional level of integration linked to monarchy.

The only temple in Mari not situated in the monumental center is the Ishtar Ush Temple. Following new studies on this sanctuary situated along the inner city wall, near a main avenue leading to the palace, I proposed that its exceptional inventory is linked to royalty, and possibly to rituals involving the royal family (Butterlin 2021b). This building may have been the place of worship of the founders of the city, buried beneath the temple in three stone graves. The link between this temple and the graves was describe by Otto (Otto 2014), but the tombs were not visible. This would have been the private sanctuary, whereas the Enceinte Sacrée was the public sanctuary, conceived for state ceremonies with a complex system of circulation and rituals. In sharp contrast to those megabuildings, the small square temples situated on top of the high terraces in Southern Mesopotamia or perhaps in Khafadge were the expression of a different relationship between power and religion.

## 4. Conclusions

The evolution of sacred architecture in Greater Mesopotamia remains a much-disputed issue and I would like to enlarge the perspective for conclusions. Since the discovery of the Göbekli buildings, the history of the temples of Greater Mesopotamia appears to be much longer than previously thought. The buildings considered the first temples, even by their excavator as temples in the garden of Eden (Schmidt 2015), have been diversely interpreted as temples by Schmidt, or as ceremonial buildings or parts of a clan system, linked possibly to ancestor worship rituals (Olivier and Notroff 2015). The first bent-axis sanctuary may be the Nevali Çori Temple, if we accept that the niche situated to the of the building room was the sacred place.

The bent-axis approach toward sacred places has largely prevailed with the development of the first tripartite temples from the sixth millennium onward in Southern Mesopotamia until the end of the fourth millennium. It was a specific arrangement suitable for welcoming ceremonies involving large groups of people. It has been possible to demonstrate that the scale of those buildings was standardized, both in Southern and Northern Mesopotamia. In Northern Mesopotamia, at least in the Tigris region, another model of tripartite architecture, with an axial entrance, was developed in the first half of the fourth millennium, but disappears with the expansion of Uruk culture in the north (Butterlin 2018, pp. 270–85).

The nature of the ceremonies performed in these ceremonial buildings remains a matter of discussion, and Uruk, and possibly the Inanna cult, played an essential role, at least during the second half of the fourth millennium BC in a global network of exchange and communication. I suggested elsewhere that a major disruption occurred at Uruk at the end of the fourth millennium, marked by the abandonment and leveling of the last monumental tripartite buildings at Eanna around Temple C, which was the center of a proto-palatial complex. It was replaced by a high terrace, and the shift is certainly indicative of major political and religious transformations (Butterlin 2018, pp. 400–4).

Regardless of happened at the end of the fourth millennium BC in Greater Mesopotamia, it is generally acknowledged that major socio-political developments occurred, leading to the development of a highly diversified system of city-states in Southern Mesopotamia and slightly later in Northern Mesopotamia. The abandonment of tripartite architecture remains to be explained since it is a major break after more the 2000 years of use in Mesopotamia. We can observe that the actual linkage between tripartite architecture and the subsequent developments in religious architecture is not a direct filiation, at least when only considering "temples".

The development of bent-axis temples both in Central Mesopotamia from the beginning of the third millennium and in Upper Mesopotamia from 2800 BC (Early Jezira 2) and the widespread use of a very diversified formula in the emerging city-states during the middle of the third millennium remain to be explained. The evolution of these ceremonial buildings in the longue durée has to be studied without hastily labeling these as "temples", for instance, which may create epistemological obstacles. The long road out of Eden presents some decisive milestones, among which are the proto-urban monumental buildings and the bent-axis shrines of the first city-states. Ristvet (2013) showed how the actual fabric of these cities was enacted through a dynamic staging of the city and its landscape. Part of that dynamic, the temples, appears as hierarchized social units, and the purpose of this paper was to show that size played a part in the staging of what Mann (1986) called the "social cages" of the early cities of Mesopotamia. This means a new method of shaping communities embedded in a network of city-states structured by economic, political, ideological, and military relations.

The scale of the cellae is but one clue in these discussions: they were part of a hierarchized staging of communities, structured along these buildings. The cella was only one part of various ceremonies that involved ritual installations (notably in the cella, but also courtyards and "chapels") and the manipulation of objects. At least in Central Mesopotamia (Diyala and Mari), statues played a major role, being the most represented objects in the inventories, among vases, masses, and incrustation panels. Tunca (2016) suggested that the statues deposited in the temples were part of an ancestor cult, and compared them to the much earlier rituals involving manipulation of skulls, statues, or figurines deposited also in votive deposits. With the development of tripartite temples, notably during the Late Chalcolithic period, this type of ritual disappears almost completely, with the only exception being the eye temple in Brak, where hundreds of little eye idols in alabaster were deposited in the terrace upon which the building stood.

The subsequent development of city-states is the result of the collapse of the Uruk system, which was clearly founded upon specific political and religious roots. It is usually considered that a continuous development occurred from the Uruk model toward

the system of city-states at the beginning of the third millennium, at least in Southern Mesopotamia. In Central and Northern Mesopotamia, new traditions developed, with new architectural models, a new method of organizing the city, not the "city" that was actually the meaning of "Uruk", but a world of cities. In this differentiated world of cities, I suggested that each temple was therefore the materialization of a part of the community involved in the urban fabric, actually represented by a people of statues. In the case of the Mari temples, the offering lists show a system of redistribution toward these temples by central institutions of various products. Among the general layout of the bent-axis type prevails a surprising diversity in terms of scale and organization of spaces. Among the differences we observed, some regional specificities appear, and they need to be explained, as do the reasons for the disappearance of this type of temple.

Along the Euphrates River, bent-axis temples were replaced by in antis temples in the middle of the third millennium in the Upper Euphrates, and at Mari after the Akkad period. During the Shakkanakû period, new temples were built upon some of the older ones, either as in antis temples or tower temples. In Diyala, this kind of temple also disappears after the Akkad period as is also the case with the widespread use of votive private statues, both in Mari and Diyala. This layout remains in the Tigridian region, notably in Ashur and Nuzi, in the first half of the second millennium. It is also at this time that two bent axis temples were built at Tell el-Daba, in the eastern Nile Delta, a singularity pointed out and discussed by Bietak (Manfred 2003). Other prevailing models of organization of temple become paramount, the langraum or breitraum models and the so-called tower temples, with the development of territorial states or empires, with their specific communication system.

Among the monumental buildings, the bent-axis type strangely survived in the layout of palaces from the third dynasty of Ur onward and, notably, during the middle Bronze Age in the classical modules of monumental amorrite blocks (either tripartite or double-halled) and later on in neo-Assyrian throne rooms.

**Supplementary Materials:** The following are available online at https://www.mdpi.com/article/10.3390/rel12080666/s1, Figure S1: Greater Mesopotamia during the middle of the third millennium, with location of bent axis temples, map by Martin Sauvage and Pascal Butterlin. Figure S2: Mari, City II temples, Mission archéologique française de Mari, Pascal Butterlin. Figure S3: Mari, Ninhursag temple, Mission archéologique française de Mari. Figure S4: Mari, cellae of city II, author's synthesis, Mission archéologique française de Mari, Pascal Butterlin.Figure S5: Mari, temple of the lord of the land, Mission archéologique française de Mari, Pascal Butterlin. Figure S6: Early Dynastic cellae in the Diyala, author's synthesis, Pascal Butterlin. Figure S7: Nippur North temple, author's synthesis, Pascal Butterlin. Figure S8: Square temples from Early Dynastic temples from southern Mesopotamia, author's synthesis, Pascal Butterlin. Figure S9: Nuzi Ashur Ishtar temple, author's synthesis, Pascal Butterlin. Figure S10: Nagar Kingdom shrines, author's synthesis, Pascal Butterlin, courtesy Marc Lebeau. Figure S11: Chronological chart of the third millennium, Pascal Butterlin. Figure S12: Favissa of the temple of the lord of the land, Mission archéologique française de Mari, Pascal Butterlin. Figure S13: Bent axis temples of Mesopotamia 1, cellae, ordered by width, author's synthesis, Pascal Butterlin. Figure S14: Bent axis temples of Mesopotamia 2, cellae, ordered by width, author's synthesis, Pascal Butterlin. Figure S15: Uruk reception rooms, author's synthesis, Pascal Butterlin. Figure S16: Mesopotamian High terraces and their environment author's synthesis, Pascal Butterlin. Figure S17: Mari, Inanna Zaza temple, circulation pattern, Pascal Butterlin Figure S18: Mari, enceinte sacrée, circulation pattern, Pascal Butterlin Figure S19: The monumental center at Khafadgé, courtesy Philippe Quenet. Figure S20: Tell Beydar, Nabada, courtesy Marc Lebeau. Figure S21: Mari, City II, monumental center, mission archéologique française de Mari, Pascal Butterlin Figure S22: Mari, monumental center, reconstruction, mission archéologique française de Mari, drawing by Françoise Laroche Tronecker and Didier Laroche.

**Funding:** This research received no external funding.

**Institutional Review Board Statement:** Not applicable.

**Informed Consent Statement:** Not applicable.

**Data Availability Statement:** Not applicable.

**Conflicts of Interest:** The author declares no conflict of interest.

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
