# Peer review of "The Long Road Out of Eden: Early Dynastic Temples, a Quantitative Approach to the Bent-Axis Shrines"

_religions, doi:10.3390/rel12080666_

Round 1

Reviewer 1 Report

This is an interesting article that tackles a long-standing trope of Mesopotamian archaeology, the meaning and use of the bent-axis ‘temple’. The data analysis is solid and the quantification approach is a good way to think through issues of capacity, access and function. But the balance between data description and analysis is uneven, with too much of the former and not enough of the latter.

General issues:

The beginning promises a lot: "…a quantitative approach, based upon assessing the capacity of those rooms, appreciated through dimension, surface, patterns of circulation, doors, and focal points. It will therefore be possible to discuss on a solid ground their accommodation capacity, as one criterion to discuss the way they were used or at least conceived….I have first looked at the dimensions of those rooms (length and width, surface) and then looked at the circulation patterns which might enlighten the way those rooms were used. This gives us some insights upon the kind of staging which was at work, a grammar of prestige and display…I put forward the idea of religiosity of movement, at different levels."

The circulation patterns, focal points, grammar of display, and religiosity of movement do not really appear in the article. The discussion focusses quite narrowly on dimensions and area sizes, with a bit on the doors. The discussion of area does have a lot of potential and is fine; in fact if all these other aspects were included, the article might be over-long.

But even if the article is reworked a bit to omit mention of circulation, focal points, etc. there are some issues to address. The emphasis on area/surface is good and could be revealing, but most of the description remains as meters squared throughout and is not related to the people who would have used these shrines, despite the author making several references to the implications for social units. This is the case for both the initial calculations of the 3rd mill shrines and for the later comparison between Uruk and 3rd mill BC buildings. The different size classes and hierarchy of sizes are certainly mathematically correct, but their meaning could be expanded much more. Eventually near the end (from bot p 17) there is some calculation of the potential capacity of these spaces in terms of people. But the equation used for this is missing and un-referenced (m2 needed per person). It seems to be based on modern capacity of rooms used for meetings, but there are huge differences in modern calculations—there are very different space requirements for people who are standing, sitting, walking, listening to music, etc. The suggestion is that benches are in use but this part is not very clear and does not seem to take unusable/unused space into account (e.g., space around an altar, space for an officiant to move around). The calculations do give some potentially useful data, in terms of some spaces being suggested as only large enough for kin groups. But whether that is plausible depends on the (missing) capacity formula used—a 30 m2 space might hold 150 people (standing) if packed to the recommended maximum of 5/m2 (see work of Keith Still [modern] and Inomata [Maya]), not only 9. Even if one end of such a room were kept open, 100 people could squeeze in. So there is a huge weakness in the argument right there. This might be resolved if more potential scenarios were calculated, e.g., seated on benches, standing, or standing and crowded. Some connection to the settlement population would also make this stronger—there is brief mention of tribal groups versus family/kin-based systems on p 18, but how do these forms of social organisation affect the size of groups that might come together for whatever events took place in these spaces? How are these defined and are we certain that tribal/kin-based are relevant in the 3rd mill BC?

Second, 2900 to 23/2200 BC is a long time in which many social and political developments occurred, including most relevantly a huge increase in urbanisation in southern Mesopotamia (number of cities) as well as increases in the size of individual cities. This would have directly affected the number of people who might need to be contained in any kind of semi-public space. Can that be seen in the S Mesop trends? In northern Mesopotamia, this period saw the shift from independent city-states to some influence (and spots of outright control) by the Akkadian kings. Did this have any impact? Changes over time are indicated here, especially for sites where there are temple sequences, but the reasons for such changes are not addressed.

Specific issues:

The first paragraph, and top two paragraphs on p 2 discussing past studies need references. There are also places in the text where an author is referred to, e.g., Evans, Porter, Frankfort, etc, but the citation (date) is missing.

The arrangement of the initial data section, from specific sites and regions, is fine, but in each case, the descriptions of room dimensions could be shorter or even eliminated from the text entirely, and placed instead in tables, where numerical data are much easier to understand (most of the numbers are in the tables, but some are not, and there is no need for such data to be in two places). Moving the numbers to tables would free up space for more discussion of meanings. The tables could also be better organised so that the rooms of similar length (or some other dimension, or class) are grouped together—now the order of the Mari table, for instance, seems to be random and one has to flip up and down the rows to compare. A column for date/level would be useful also. There are similar issues for the Diyala table, which is organised by building but could be reorganised to reflect the classes identified. For the Nippur North Temple table, what are the two different measurements for length?

For all the tables, I recommend that more data are included and that the same data categories are included in each (there is variability in number of columns from table to table), for comparability. So add columns for Area (where not already included), Length-Width Ratio (where not included), Date, Number of Doors.

What does the width:length ratio tell us? This figure is calculated but not really discussed in terms of implications. There is some discussion of width, related to beam length (pp. 16-17), but it is not linked well to the ratio, area and capacity. Maybe consider omitting this?

Sometimes the doors (number, location) are mentioned in the text, but this information is not taken forward (as above, this should be a column in all the tables, with discussion of circulation and access, or omit it entirely). Frankfort & Delougaz’s identification of the change between a through-room with a specific focal point to an end-point is identified as important (pp 7, 13), but more could be done with the idea of a route moving through and the lived experience of the buildings’ users.

In the discussion at the bottom of p 13, the author’s meaning is not quite clear—do they agree that the change in plan is related to a new concept of the sacred? Or a new relationship of worshippers to the sacred? The top of p 14 brings in the idea of constant renewal of temples within the constant process of settlement change, but it is not clear where this is going. The question of whether ‘temples’ really were temples is raised on p 14, via the Chuera statues as evidence of an ancestor cult, and the problem that temples are often identified as such through inference from single inscriptions. But again, there is no conclusion drawn from this.

Author Response

Thanks for the comments and corrections, table have been corrected and treated the same way,  as the citations. 

Additional comments have been provide in part 3.3, about circulation patterns, doors, and in part 3.4 to discuss the actual size of the communities involved. 

The calculation reference is also added as is explained the way people are arranged in this type of arrangement. 

as for the  discussion about ancestor cult, it is linked to the interpretation of statues,  not dimensions, that would need another paper to be discussed in depth. 

Reviewer 2 Report

This paper offers important data, and the value of the regional and chronological comparisons is significant. However it needs considerable work in two areas. The first is its English. It is difficult to read and at a couple of points, just does not make sense. The second is the various lacunae between data and discussion. As it stands, I am really not sure what the author is arguing is the outcome of the study.

For example, a critical question posed at the beginning of the paper never really gets addressed “One of the main questions is to understand why tripartite architecture which was the matrix of the development of monumental architecture in Mesopotamia ceased almost completely to be the major pattern of organization of prestige architecture." At the end of the paper the author proposes that size differentials may be related to the size and nature of the social units using the buildings. Although I think this is a very important idea, there is no clear linkage between size and social structure provided on theoretical or empirical terms. Accommodation capacity can be result of several different factors, such as the nature of particular religious rituals - not all are intended for the public - or beliefs associated with specific gods. So this part of the paper needs to be developed much more explicitly. And then the question posed at the outset needs to be addressed, because if it holds, this shift from tripartite structures to other architecture forms has major socio-political implications. Does it mean there is a shift in social organization across this time?  If so it would be counter intuitive to most scholars, and thus important to make a convincing argument. Or is that there was always regional distinctions in social terms? Please clarify as this is an important issue. There is one point where it seems that the main explanation for size differentials is the availability of suitable trees for roof beams. 

More specifically, the author contrasts family or kin-based temple structures (small) with tribal ones (larger). Both tribes and families are kin-based, just at different scales. And, of course, there are different scales of families too. There needs to be a more extensive discussion of what these social units are, how they work. How does this idea of social units, different type of religious foundations relate to the statement in the first paragraph that "that the roots of power lay in the sacred, that kings were first of all “king priests” and that the birth of the state was actually linked to the institutionalization of religion?" The possibilities are provocative, but they are not picked up and developed.

The connection to Gobekli Tepe at the end of the paper is spurious as it stands – there is no discussion for example at the beginning of the paper as to any potential relationship between Neolithic and late chalcolithic/ED religious structures. And it is my understanding that Schmidt, and others of the team, understand these structures not just as temples, but the pillars within them as actuals gods, and not ancestors. Not that I agree with that interpretation, but the point is, the case has to be made for one idea or the other. Then at the very end,  Michael Mann's idea of "social cages" is introduced with no explanation as to what that idea is, or its connection to the temples of greater Mesopotamia.

This will be a very important paper, if these kinds of concerns are addressed.

Author Response

Thanks for the comments

I provide a new version discussing more in depth the question of the transition between fourth millennium tripartite architecture and the third millennium. the question has been already discussed in (removed for peer review). 

I provide an extensive discussion upon the link between the different social groups possibly linked to temples and kingship, at least in the case of Mari , see 3.3 and 3.4 corrected. 

as for Göbekli, the interpretation has evolved considerably and the question of the existence of a clanic system and possibly an ancestor cult has been raised. The concentration is one specific place of organic structures places side by side, with similar patterns but also clear specificities reminds the similar situation observed at Eanna in Uruk or later with the temples at Beydar or Mari for instance. 

Tunca has even suggested a link between skull rituals, votive deposits of statues as attested until the Halaf and the votive deposits of statues in early Dynastic as expression of an ancestor cult, but this is largely beyond the scope of the paper, to my sense. 

Round 2

Reviewer 1 Report

The article is much improved and the author(s) have addressed my concerns with the clearer presentation of data and the support for their interpretations. The article needs proof-reading and copy-editing to improve the English but is otherwise a strong contribution.

Author Response

Thank you, English is improved. 

Reviewer 2 Report

The paper is a clear presentation of the evidence and highlights the significance of the problems. There are still some minor corrections to the English required, such as line 31 "has prevailed the idea of". This should be "the idea has prevailed that" or "the idea that.... has prevailed". This is just one example. 

Author Response

thank you corrections included !